# Detection of the thietane precursor in the UVA formation of the DNA 6-4 photoadduct

Luis A. Ortiz-Rodríguez [1], Christian Reichardt[1,3], Sean J. Hoehn [1,3], Steffen Jockusch [2] & Carlos E. Crespo-Hernández [1✉]

Notwithstanding the central biological role of the (6-4) photoadduct in the induction of skin cancer by sunlight, crucial mechanistic details about its formation have evaded characterization despite efforts spanning more than half a century. 4-Thiothymidine (4tT) has been widely used as an important model system to study its mechanism of formation, but the excited-state precursor, the intermediate species, and the time scale leading to the formation of the (6-4) photoadduct have remained elusive. Herein, steady-state and time-resolved spectroscopic techniques are combined with new and reported quantum-chemical calculations to demonstrate the excited state leading to the formation of the thietane intermediate, its rate, and the formation of the (6-4) photoadduct using the 5'-TT(4tT)T(4tT)TT-3' DNA oligonucleotide. Efficient, sub-1 ps intersystem crossing leads to the population of a triplet minimum of the thietane intermediate in as short as 3 ps, which intersystem crosses to its ground state and rearranges to form the (6-4) photoadduct.

[1] Department of Chemistry, Case Western Reserve University, Cleveland, OH 44106, USA. [2] Department of Chemistry, Columbia University, New York, NY 10027, USA. [3] These authors contributed equally: Christian Reichardt, Sean J. Hoehn. ✉email: carlos.crespo@case.edu

The *cis*-syn cyclobutane dimers and the (6–4) pyrimidine–pyrimidinone adducts are the most frequent photolesions occurring at bipyrimidine sites in DNA of all forms of life[1,2] following exposure to ultraviolet (UV) solar radiation[1,3–6]. In particular, the (6–4) photoadducts hold a pyri-midinone chromophore within, which can act as an internal photosensitizer to form additional cyclobutane dimers in DNA[7,8] or can absorb UVA/UVB radiation to form Dewar valence isomers[1,2,9].

Despite more than six decades of focused effort in the field, there is still substantial debate regarding the excited state, leading to the formation of the (6–4) photoadducts, as well as, a complete lack of information regarding the timescales at which the oxetane, azetidine, and thietane intermediates are formed[1,2,10]. Indeed, as depicted in Fig. 1 for the oxetane intermediate, the $^1n\pi^*$, $^3n\pi^*$, $^3\pi\pi^*$, and charge-transfer singlet and triplet states have all been proposed as precursor states in the formation of these elusive intermediates[2,4,5,10–27]. Early investigations suggested that the (6–4) photoadducts may have originated from excited singlet states because triplet photosensitization did not lead to the formation of these photoadducts, and triplet quenchers did not prevent their formation[3–5,28]. Recent theoretical investigations have presented conflicting predictions with several groups providing evidence for the participation of excited triplet states in the formation of the oxetane and azetidine intermediates[14,15,18,21], while others substantiate participation of charge-transfer states of either triplet or singlet multiplicity[12,17,19,29–32]. The participation of an electron transfer pathway in the formation of the oxetane intermediate was proposed recently using experimental and computational investigations for a thymine–thymine dinucleotide analog in which the 5′-thymine was replaced by a meta-xylene moiety[33]. This work demonstrates that 5′ → 3′ electron transfer is a necessary first step in the formation of the xylene-thymine (6–4) photoadduct analog. However, the authors were unable to experimentally detect the oxetane intermediate[33], and only computational predictions supported its participation in the formation of the (6–4) photoadduct in the xylene-thymine dinucleotide. Furthermore, the redox properties of meta-xylene are significantly different from those of thymine, favoring a complete electron transfer pathway in this dinucleotide analog[33] relative to a partial charge-transfer pathway thymine–thymine dinucleotide[18–20].

The 4-thiothymidine–thymine dinucleotide is an important model system that has been widely used to study and understand the mechanism of formation of the (6–4) photoadduct in DNA[27,34–38]. Furthermore, to the best of our knowledge, thietane is the only intermediate that has been characterized spectroscopically at room temperature[34]. This dinucleotide analogue is based on the photochemistry of 4-thiothymidine (4tT), in which the oxygen atom at the C4 position of the 3′-thymine moiety is substituted by a sulfur atom. This single-atom substitution shifts the absorption spectrum of thymidine to lower energies, allowing for the selective excitation of 4tT with UVA radiation when incorporated in DNA dinucleotide and oligonucleotide systems[34,37]. Importantly, UVA excitation of 4tT-containing oligonucleotides results in the formation of the S5-(6–4) photoadduct analog of the natural (6–4) photoadduct, without the competitive formation of the [2 + 2] cyclobutane thymidine dimer observed in DNA[34,37]. However, despite of the wide use of the thietane intermediate as a stable substrate analog of the oxetane intermediate in DNA[34,36,37,39] and RNA[25,26,40,41] systems, crucial mechanistic information is still lacking, such as the excited-state precursor and the timescale leading to its formation.

In this contribution, steady-state absorption and emission spectroscopic techniques, time-resolved luminescence spectroscopy, femtosecond broadband transient absorption, and time-dependent density functional (TDDFT) calculations are combined to reveal the excited state leading to the formation of the thietane intermediate, its rate of formation, as well as the subsequent formation of the (6–4) photoadduct in DNA. A 7-mer single-stranded DNA oligonucleotide containing two pairs of adjacent 4-thiothymidine–thymidine nucleotides (4tTT), 5′-TT(4tT)T(4tT) TT-3′, referred to as $T(T4tT)_2T_2$, hereafter, is investigated in phosphate buffered saline (PBS) solution at pH 7.4. For comparison, consecutive experiments are performed using 4tT nucleoside under equal experimental conditions, because its photophysics and excited-state dynamics are now well established[42–51]. It is shown that efficient, sub-1 ps intersystem crossing occurs in this single-stranded DNA oligonucleotide, leading to the population of a triplet-state minimum of the thietane intermediate in a time delay as short as 3 ps ($\tau = 290 \pm 70$ ps) in $25 \pm 5\%$ yield from two separate measurements. The triplet-state minimum of the thietane intersystem crosses to the ground state of the thietane intermediate and then rearranges to form the characteristic (6–4) photoadduct. A fraction of the triplet-state population in the oligonucleotide that is not conducive to (6–4) photoadduct formation, decays back to the ground state in ca. 23 ns—two order of magnitude faster that triplet decay in the 4tT monomer[42]. Furthermore, it is shown that the quantum yield of singlet oxygen generation in $T(T4tT)_2T_2$ is 88% lower than in the 4tT monomer under equal experimental conditions.

## Results

**Steady-state absorption and emission spectra.** Figure 2 shows the steady-state absorption and emission spectra of $T(T4tT)_2T_2$ and 4tT in PBS pH 7.4 at room temperature. Both $T(T4tT)_2T_2$ and 4tT exhibit a low-energy absorption band with maximum at

**Fig. 1 Literature-proposed excited-state precursors of the oxetane intermediate**[2–5,10–21,28]. 5′-TpT-3′ represents a thymine–thymine DNA dinucleotide, while 6-4PP represents the final (6–4) pyrimidine–pyrimidinone adduct formed after DNA absorbs ultraviolet radiation. The 5′-TpT-3′ structure was chosen as a representative DNA structure for simplicity with the purpose of highlighting that two thymidine bases need to be close to each other for the reaction to occur. This figure does not imply that the reaction is limited to a dinucleotide, but it can also occur in single- and double-stranded oligonucleotides. The oxygen atom involved in the initial [2 + 2] cycloaddition reaction is highlighted with red color. CT stands for charge transfer.

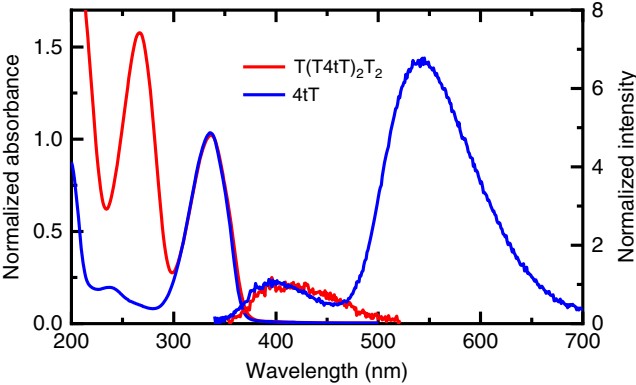

**Fig. 2 Normalized absorption and emission spectra for 4tT and T (T4tT)₂T₂.** Room-temperature absorption and emission spectra for 4tT in PBS at pH 7.4 are shown using blue lines, while those for T(T4tT)₂T₂ are depicted using red lines. We remark that the emission spectrum for T(T4tT)₂T₂ has a very small intensity and may not be measured accurately within the sensitivity of the spectrofluorometer used in this work. Photoexcitation was performed at 340 nm.

ca. 336 nm, which is slightly red shifted in T(T4tT)₂T₂ compared with 4tT. In addition, the absorption spectrum of T(T4tT)₂T₂ shows a high-energy band centered at 266 nm, whereas the 4tT monomer shows a similar band at 237 nm that has significantly lower intensity. The band at 266 nm is primarily due to the absorption by the thymidine nucleotides in the T(T4tT)₂T₂ oligonucleotide, whereas it is due to a $S_0 \rightarrow S_n$ transition in the 4tT monomer[48]. Intriguingly, while both fluorescence (~400 nm) and phosphorescence (~540 nm) emission are observed for the 4tT monomer in aqueous solution (Fig. 2)[44], no detectable phosphorescence is observed for T(T4tT)₂T₂, and only a negligibly small fluorescence-emission band is observed, which could not be accurately measured within the sensitivity of the spectrofluorometer used. The lack of phosphorescence emission in the oligonucleotide suggests a significant quenching of the near-unity triplet yield reported for the 4tT monomer[42,44] in solution that occurs when 4tT is incorporated in the single-stranded DNA oligonucleotide.

**Determination of quantum yields of singlet-oxygen generation.** To further investigate the proposed triplet quenching of 4tT in T(T4tT)₂T₂, the nglet-oxygen quantum yield of both 4tT and T(T4tT)₂T₂ were measured under equal conditions using time-resolved phosphorescence spectroscopy. The singlet-oxygen phosphorescence signals were obtained in deuterated TRIS buffer solutions at pH 7.4, following excitation at 355 nm. As shown in Fig. 3, a significant decrease in the intensity of the singlet-oxygen phosphorescence signal is observed for T(T4tT)₂T₂ under both O₂ and air-saturated conditions compared with 4tT under air-saturated conditions. Indeed, nearly an order of magnitude reduction in singlet-oxygen quantum yield was measured for T-(T4tT)₂T₂ relative to 4tT (Table 1). The 88% decrease in the singlet-oxygen yield lend further support to the idea that the incorporation of 4tT in DNA results in a significant quenching of the excited triplet-state population.

**Ultrafast triplet-state dynamics.** Femtosecond broadband transient-absorption spectroscopy was used to directly probe the excited-states dynamics of both 4tT and T(T4tT)₂T₂ following excitation at 342 nm in PBS solutions at pH 7.4. Figure 4 compares the transient-absorption spectra of both 4tT and T(T4tT)₂T₂, which were measured consecutively under equal experimental conditions. The multidimensional transient data

were globally fitted using a four-component sequential kinetic model, where the fourth kinetic component required a large lifetime to satisfactorily fit the long-lived transient signals of both T(T4tT)₂T₂ and 4tT. As can be seen in Fig. 4, the long-lived transient signal of both T(T4tT)₂T₂ and 4tT last beyond the 3 ns time window of our femtosecond spectrometer. Representative kinetic traces and best global fit curves are shown in Fig. 5, while the associated lifetimes are collected in Table 1. The evolution-associated difference spectra (EADS) extracted from global and target analyses of the transient data for the monomer and the single-stranded oligonucleotide are shown in Supplementary Figs. 1 and 2, respectively. The initial lifetime for the 4tT nucleoside is in good agreement with that reported earlier[44,48]. The second and third lifetimes of 4tT are associated to vibrational relaxation in the triplet state and fractional triplet self-quenching due to aggregate formation[44,45,48]. Hence, their magnitudes are expected to depend on both the ground-state concentration of 4tT and the experimental conditions used.

## Discussion

As can be seeing on the top panel of Fig. 4, the initial ca. sub-1 ps dynamics for both the monomer and the single-stranded DNA oligonucleotide are essentially the same. The dynamics are characterized by three primary bands—two with negative amplitude (<365 nm and from ca. 365 to 500 nm) and one with positive amplitude (ca. > 500 nm)—the positive-amplitude band in the visible rising and shifting to the blue within a 1 ps time delay (Fig. 4a, d). These bands are assigned to ground-state depopulation, stimulated emission, and sub-1 ps intersystem crossing leading to the population of the lowest-energy triplet state ($T_1$), respectively, in agreement with previous experimental observations for the 4tT monomer[43–45,48]. Significantly, the femtosecond transient-absorption experiments demonstrate that ultrafast intersystem crossing dynamics to populate the $T_1$ state also occurs in the single-stranded DNA containing 4tT. To the best of our knowledge, this is the first experimental evidence of ultrafast (i.e., within $\tau_1 = 0.26 \pm 0.02$ ps) and efficient triplet-state population reported for a single-stranded DNA oligonucleotide. This experimental result is fully supported by high-level quantum-chemical calculations performed recently for a double-stranded DNA oligonucleotide containing 4tT[23], which predictions are discussed in more detail below.

Next, we devote particular attention to both the dynamics and the spectral changes observed in the triplet absorption spectrum from ca. 380 to 670 nm. Panels b and e in Fig. 4 show the initial decay dynamics of the $T_1$ state of 4tT as a monomer and when incorporated adjacent to thymine bases in the single-stranded DNA, respectively. Significant spectral and dynamical differences are observed between 4tT and T(T4tT)₂T₂ during the time window from ca. 1 to 65 ps and beyond, specifically in the spectral region from ca. 380 to 500 nm. While the intensity in the spectral region from 380 to 450 nm decreases with an increase in time delay for 4tT, the intensity in the spectral region from 380 to 495 nm actually increases for T(T4tT)₂T₂ (Fig. 4b, e). Furthermore, an apparent isosbestic point is observed at ca. 495 nm in the oligonucleotide (not observed in the monomer), which is indicative of a state-to-state process. Conversely, the intensity of the lower-energy triplet band at 565 nm shifts to 550 nm and slightly decreases during the time window between ca. 1 to 5 ps in both samples. Once the triplet band shits to 550 nm, it continues to decay in both the monomer and the single-stranded oligonucleotide. The result in Fig. 4c that the triplet state of 4tT continues to decay monotonically in hundreds of nanoseconds, is in good agreement with previous observations[42,50]. Supplementary Fig. 3 shows

that the dynamics of the transient-absorption bands ~430 and 550 nm are quite different in the monomer compared to the single-stranded DNA oligonucleotide. The blue shift observed in the transient spectra during the ca. 1 to 5 ps time window is associated with vibrational relaxation in the triplet state of 4tT in both the monomer and the single-stranded oligonucleotide ($\tau_2$ in Table 1). In addition, a small fraction (ca. 30 ± 5%) of the triplet-state population in the 4tT monomer decays by triplet self-quenching[44,45] to the ground state with a lifetime of 56 ± 8 ps under the experimental conditions used.

Returning to the dynamics in the single-stranded DNA oligonucleotide, the data presented in panels e and f of Fig. 4 demonstrate that at least two different transient species are participating in the dynamics during the time window from ca. 1 ps to 3 ns. While the transient spectra in the spectral region from ca. 380 to 460 nm continues to growth for up to ca. 1 ns, the transient-absorption band with maximum at 550 nm decays during the same time window. On the one hand, we assign the decrease of the 550 nm absorption band as due to a large fraction of the triplet state of 4tT in the oligonucleotide decaying non-radiatively to the ground state, which we estimate to occur in ca. 23 ns ($\tau_4$), assuming triplet self-quenching of 4tT with adjacent thymine bases that is not conducive to reaction. Absorption spectra recorded before and after the laser experiments support the idea that a large fraction of the photoexcited 4tT in the oligonucleotide decays back to the ground state (Supplementary Fig. 4a). On the other hand, the transient species increasing in the spectral region from ca. 380 to 460 nm is assigned to the population of a triplet-state minimum of the thietane intermediate (Fig. 6), which is populated from the interaction of the triplet

state of 4tT with the adjacent 5′-thymine base in the oligonucleotide. Using Eq. 1 and the results from the global fit analysis, a relative quantum yield of 25 ± 5% was estimated from two separate measurements for the population of the triplet-state minimum of the thietane intermediate. This value is larger than the quantum yield reported by Warren et al.[37] of 10% for the thietane intermediate in a double-stranded oligonucleotide containing one 4tT. This suggests that a fraction of the triplet-state minimum of the thietane intersystem crosses to populate the thietane intermediate in the ground state (Fig. 6), while another fraction is not conducive to the formation of the thietane intermediate in the ground state. This result is not necessarily surprising because the 4tT in the single-stranded oligonucleotide can explore a larger conformational space (such as unstacked and stacked conformations) than when it is incorporated in double-stranded oligonucleotides.

Interestingly, the amplitude of the triplet-state minimum of the thietane at 430 nm increases from a time delay of ca. 3 ps to up to 1 ns (see Fig. 5b). This suggests that the triplet minimum of the thietane begins to populate in ca. 3 ps, while intersystem crossing from the triplet minimum of the thietane to the thietane intermediate in the ground state occurs on a longer timescale than 3 ns. Attempts to measure the rate of intersystem crossing to the thietane intermediate in the ground state were unsuccessful because of the limited white light generation below 420 nm of the nanosecond transient-absorption spectrometer available to us[53]. This is consistent with TDDFT calculations reported in the Supplementary Information, which suggest that the thietane intermediate in the ground state absorbs at wavelengths shorter than 300 nm (Supplementary Fig. 5). Similarly, we were unable to measure the intersystem crossing from the triplet minimum of the thietane to the thietane intermediate in the ground state, nor the triplet decay lifetime of 4tT in the oligonucleotide, because of the low sensitivity of our nanosecond spectrometer[53,54]. While this is a disappointing result, it lends further support to the idea that most, if not all, of the triplet state of 4tT in the oligonucleotide that is not conducive to thietane formation decays to the ground state in tens of nanoseconds, as the global and target analysis of the data suggest. A probable quenching mechanism is triplet self-quenching with an adjacent thymine base.

Recent CASPT2 quantum-chemical and molecular dynamics simulations that include explicit solvent effects have been performed[23], which further support the assignment of the transient species in the spectral region from 380 to 460 nm to the triplet minimum of a thietane intermediate (see Fig. 6). Specifically, Xie and Cui presented a theoretical investigation that provides crucial mechanistic information about the photochemical reaction of the T4tT bipyrimidine sequence in a double-stranded 5′-ACCT4tT CGC-3′•3′-TGGAAGCG-5′ DNA oligonucleotide[23]. The authors identified five efficient nonadiabatic relaxation pathways starting from the initially populated $^1\pi\pi^*$ (S$_2$) state of the T4tT sequence, which populate the T$_1$ state of 4tT. These nonadiabatic decay pathways involve two- and four-state intersection crossings, which the authors labeled as S$_2$/S$_1$ and S$_2$/T$_2$/S$_1$/T$_1$, respectively[23],

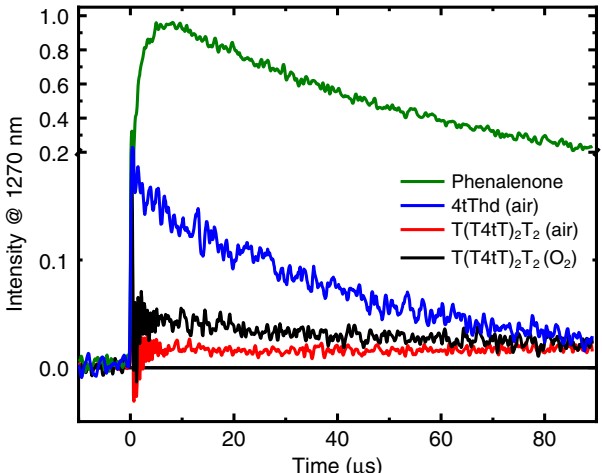

**Fig. 3 Time-resolved phosphorescence signals at 1270 nm.** Singlet-oxygen decay traces were generated by pulsed photoexcitation (355 nm, 7 ns pulses) of 4tT (blue line), T(T4tT)$_2$T$_2$ (red and black lines), and phenalenone (green line, $\Phi_\Delta = 0.98$)[52] in O$_2$ (black line) and air-saturated (blue and red lines) TRIS-buffered D$_2$O solution.

**Table 1 Transient-absorption lifetimes and quantum yields of singlet-oxygen generation.**

| DNA system | $\tau_1$ (ps)[a] | $\tau_2$ (ps)[a] | $\tau_3$ (ps)[a] | $\tau_4$ (ns) | $\Phi_\Delta$ (air)[b] | $\Phi_\Delta$ (O$_2$)[b] |
|---|---|---|---|---|---|---|
| T(T4tT)$_2$T$_2$ | 0.26 ± 0.02 | 3 ± 2[c] | 290 ± 70[c] | 23[a,d] | 0.02 ± 0.01 | 0.04 ± 0.01 |
| 4tT | 0.24 ± 0.02 | 2 ± 1[c] | 56 ± 8[c] | 4200 ± 300[e] | 0.17 ± 0.02 | – |

[a]Measured in PBS pH 7.4 following excitation at 342 nm with reported errors equal to twice the standard deviation of at least two separate measurements.
[b]Measured in deuterated TRIS buffer solutions at pH 7.4 following excitation at 355 nm.
[c]$\tau_2$ and $\tau_3$ are correlated to each other, and the reported large errors take into consideration this correlation.
[d]Extrapolated lifetime assuming a complete exponential decay to the ground state.
[e]Triplet lifetime calculated at infinite dilution in acetonitrile[42].

in good agreement with analogous calculations done for the 4tT monomer in solution[48]. Furthermore, the calculations predicted high-energy barriers between both the $S_1$ state and the ground state ($S_0$), and between the $T_1$ and the $S_0$ states[23]. The results shown in Fig. 4 are in agreement with such high energy barriers, evidencing ultrafast and efficient population of the triplet state of 4tT in high yield in the $T(T4tT)_2T_2$ single-stranded DNA oligonucleotide. This hypothesis is further supported by the prediction

of large spin–orbit couplings between the $S_1/T_1$ and $S_1/T_2$ crossing structures of 94.1 and 69.2 cm$^{-1}$, respectively[23], leading to the efficient and ultrafast population of the $T_1$ state of 4tT through a $T_2/T_1$ conical intersection in the double-stranded DNA oligonucleotide.

After reaching the $T_1$ state of 4tT in the oligonucleotide, the calculations predict that a [2 + 2] cycloaddition reaction takes place leading to the population of a thietane triplet minimum,

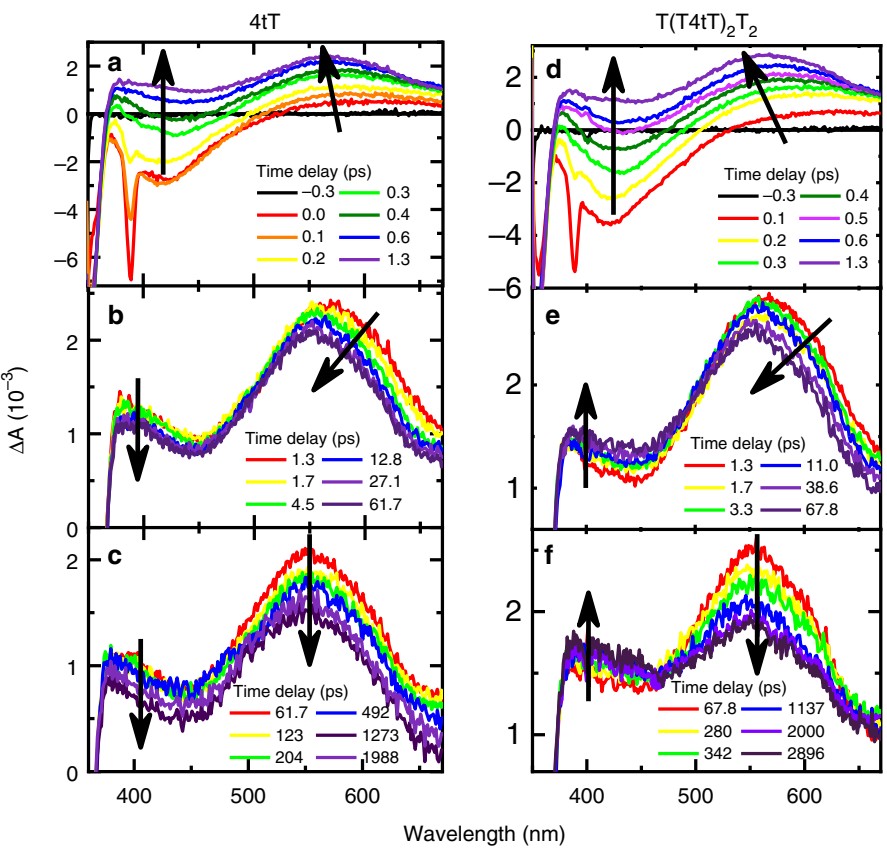

**Fig. 4 Broadband transient-absorption spectra for 4tT and T(T4tT)₂T₂.** Left panel (i.e., **a–c**) depicts the transient data for the 4tT nucleoside, while the right panel (i.e., **d–f**) shows the transient data for $T(T4tT)_2T_2$ both in PBS pH 7.4 upon excitation at 342 nm. Stimulated Raman emission from the water solvent is observed at 385 nm within the cross-correlation of the pump and probe beams in both data sets (i.e., panels **a** and **d**), and its maximum amplitude was used to define the time delay equal to zero picoseconds. The transient-absorption spectra are represented with solid lines using a rainbow of colors. Black arrows indicate the transition of the absorption bands as a function of time delay and are used as a guide to the eye.

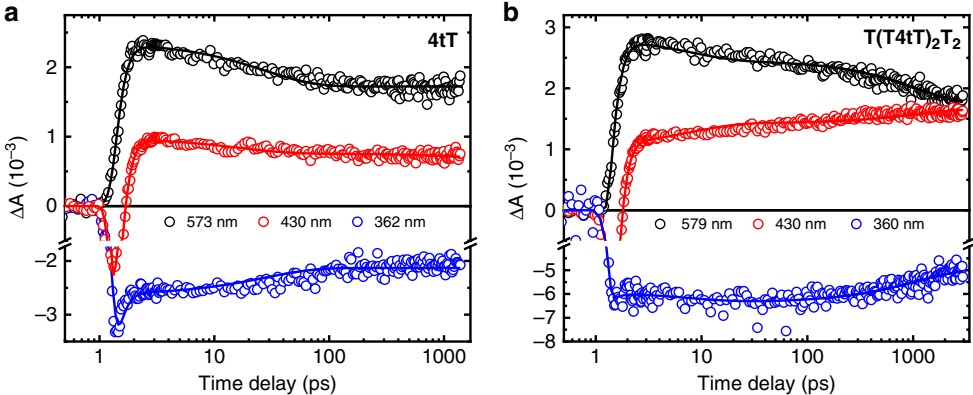

**Fig. 5 Representative kinetic traces and fits at three probe wavelengths.** Panel **a** shows kinetic traces for 4tT at the probe wavelengths of 573 nm (black), 430 nm (red), and 362 nm (blue), while panel **b** depicts kinetic traces for T(T4tT)₂T₂ at the probe wavelengths of 579 nm (black), 430 nm (red), and 360 nm (blue). Both samples were dissolved in PBS solvent at pH 7.4 and excited at 340 nm. The raw data are represented with open circles, while the lines represent the best fits obtained from global and target analyses to the multidimensional transient-absorption data.

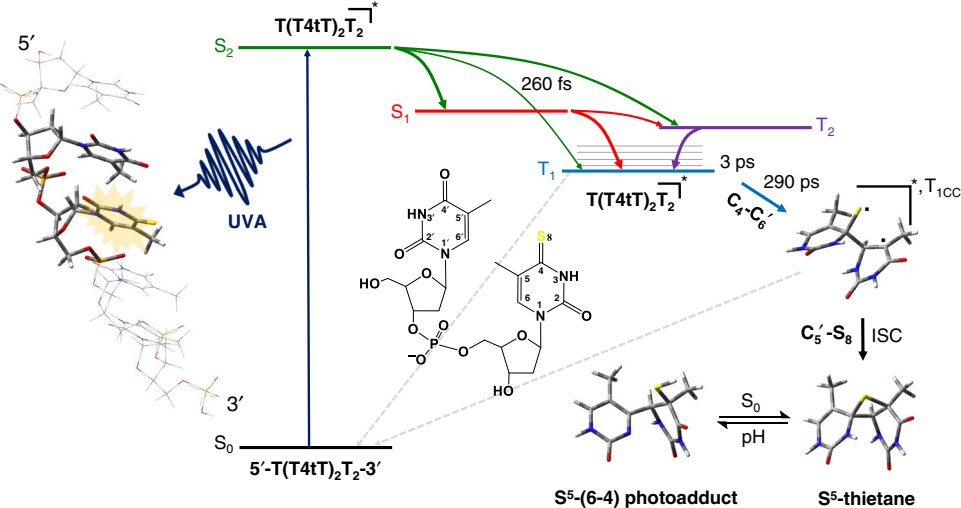

**Fig. 6 Proposed mechanism for the formation of the DNA (6–4) photoadduct in PBS at pH 7.4.** The colored arrows represent electronic transitions. A 4tT base is highlighted with light orange in the single-stranded DNA sequence depicted to the left to imply that 4tT bases are selectively photoactivated by the UVA radiation. 4tT and the thymidine 5′ relative to the 4tT are represented as tube in the single-stranded DNA sequence, while the other nucleobases are depicted as wireframe in order to highlight the importance of having a thymine 5′ relative to the 4tT for the reaction to occur. In the chemical structures yellow, red, blue, gray, and white, represent sulfur, oxygen, nitrogen, carbon, and hydrogen, respectively.

labeled $T_{1CC}$, where the subscript CC refers to the formation of a covalent bond between C4–C6′ atoms (see Fig. 6)[23]. Note that the C5′–S8 bond of $T_{1CC}$ remains broken, but with a weak interaction represented in Fig. 6 by two unpaired electrons. A $S_0/T_{1cc}$ crossing point was calculated at the QM(CASSCF)//MM/6-31 G* level with a spin–orbit coupling constant of 32.6 cm$^{-1}$, suggesting that this intersystem crossing pathway plays an important role in the subsequent [2 + 2] cycloaddition reaction. According to Xie and Cui[23], this reaction pathway can be divided into two phases. The first phase is a stepwise and adiabatic cycloaddition in the triplet state, while the second phase is a stepwise and non-adiabatic process between the triplet and the ground state of the thietane intermediate.

Using the nuclear coordinates reported by the authors[23], we have calculated the absorption spectrum of the triplet minimum of the thietane ($T_{1CC}$) using TDDFT and including a reaction-field solvation model (IEFPCM)[55] to take into consideration the dielectric effect of water in the vertical excitation energies. As shown in Supplementary Fig. 6, the absorption spectrum of the triplet minimum of the thietane intermediate exhibits two absorption bands with maxima around 362 and 442 nm. The calculated absorption spectrum is in excellent agreement with the assignment of the increase in amplitude of the transient-absorption band in the spectral region from ca. 380 to 495 nm of panel e and f of Fig. 4 to the population of a triplet minimum of the thietane intermediate. The vertical excitation energies, the character of the electronic transitions, and their oscillator strengths are reported in Supplementary Table 1, whereas the Kohn–Sham orbitals are shown in Supplementary Fig. 7.

Supplementary Figs. 8 and 9 compare the normalized decay traces for 4tT and T(T4tT)$_2$T$_2$, respectively, at 430 and 550 nm probe wavelengths. As can be seen in those figures, there is a clear rise in the amplitude for T(T4tT)$_2$T$_2$ at 430 nm not observed in the monomer, while the triplet state of 4tT in the oligonucleotide at 550 nm decays with a faster lifetime than the triplet state of the 4tT monomer. These results lend support to the proposal that the absorption spectrum of the triplet minimum of the thietane overlaps with the absorption spectrum of a residual population of the triplet state of 4tT in the single-stranded oligonucleotide in the spectral region from ca. 380 to 495 nm.

The question now arises as to how exactly the thietane intermediate is formed in the ground state? According to the calculations, there is a large energy barrier of ca. 30 kcal mol$^{-1}$ to form the C5′–S8 bond from the triplet minimum of the thietane[23]. As mentioned above, this is consistent with the continuous increase in the amplitude of the transient-absorption signal at 430 nm observed Fig. 5b, suggesting that the $T_{1CC}$ decays on a time delay longer than 3 ns. Indeed, the calculations predict that the triplet state of the thietane intersystem crosses to the ground state near the $S_0/T_{1cc}$ intersection (spin–orbit coupling of 32.6 cm$^{-1}$) to complete the cycloaddition reaction, as depicted in Fig. 6. The calculations predict a barrierless formation of the C5′–S8 bond in the ground state, leading to the thietane formation[23]. The thietane intermediate in the ground state then rearranges through a concerted process with the simultaneous fission of the C4–S8 bond and the formation of the S8–H9 bond, leading to the formation of the (6–4) photoadduct[23]. The calculations suggest that the formation of the (6–4) photoadduct occurs with a relatively small energy barrier of 22 kcal mol$^{-1}$ in the ground state[23].

To provide experimental evidence that the (6–4) photoadduct is formed in T(T4tT)$_2$T$_2$, we recorded the steady-state absorption and emission spectra for the T(T4tT)$_2$T$_2$ oligonucleotide before and after laser irradiation (Supplementary Fig. 4a), under equal conditions used for the transient-absorption experiments. The oligonucleotide was irradiated for 35 min at 342 nm, equivalent to a 202 J cm$^{-2}$ irradiation dose. After normalizing the absorption spectra at 334 nm, where 4tT has its absorption maximum, the difference absorption spectrum reveals an absorption shoulder with maximum around 320 nm, which is characteristic of the 5-methyl-2-pyrimidone moiety in the (6–4) photoadduct[2,7,8,27,37]. The difference spectrum is also in good agreement with the absorption spectrum of the (6–4) photoadduct reported by Warren et al.[37] in a double-stranded DNA oligonucleotide containing the T4tT sequence. Furthermore, excitation of the irradiated solution of the T(T4tT)$_2$T$_2$ oligonucleotide at 320 nm results in the emission spectrum shown in Supplementary Fig. 4c. The emission spectrum shows a band maximum at 390 nm, which is consistent with the emission spectrum of (6–4) photoadduct reported previously[37]. In addition, the excitation spectrum collected near the maximum of the emission band at 390 nm is

shown in Supplementary Fig. 4b. Within experimental uncertainties, it agrees with the shoulder observed in the difference absorption spectrum, as well as, with the absorption spectrum of (6–4) photoadduct[37], supporting the idea that this photoadduct is formed under the experimental conditions used in our study.

Previous investigations have shown that the thietane intermediate is in equilibrium with the (6–4) photoadduct (Fig. 6)[34,37], the former being the predominant species at neutral pH. Importantly, the yield of the (6–4) photoadduct increases under basic conditions[34]. Hence, to provide further evidence for the formation of the (6–4) photoadduct in the T(T4tT)$_2$T$_2$ oligonucleotide, we prepared buffered solutions of equal concentrations at pH of 6.2, 7.4, and 8.2 using the irradiated solution of the oligonucleotide at pH 7.4, and measured their emission and excitation spectra. As shown in Supplementary Fig. 10, the emission spectrum increases with an increase in pH, while the excitation spectrum of all three solutions exhibits a maximum at 320 nm, in good agreement with the absorption spectrum of (6–4) photoadduct[37]. Collectively, the experimental results reported in this study and the QM(CASPT2//CASSCF)/MM calculations reported by Xie and Cui[23] demonstrate that the thietane intermediate, which originates from the initial triplet state population of 4tT in the single-stranded DNA oligonucleotide, is the key intermediate in the formation of (6–4) photoadduct. Therefore, our experimental results support the idea that the (6–4) photoadduct may be formed from the triplet state between two adjacent thymine or uracil bases[14,15,18,21], if the excited triplet state of thymine or uracil is populated upon UV excitation in natural DNA or RNA, respectively. Indeed, recent experimental results provide evidence of the population of the triplet state in both thymine and uracil monomers[56–62], as well as in single-stranded thymine oligonucleotides[2,59,63], upon excitation at 266 nm. The low triplet yields reported for these natural systems may explain why CPD are formed in higher yields than (6–4) photoadducts[3].

Previous experimental results demonstrate that 4tT acts as an effective triplet-energy transfer photosensitizer to molecular oxygen when photoactivated in its monomeric form, which was supported by the high quantum yield of singlet-oxygen generation of 42% under O$_2$-saturated acetonitrile solution[46,50]. However, indirect measurements of the generation of reactive oxygen species have suggested that when 4tT is incorporated into DNA, negligible amounts of reactive oxygen species are generated[27]. In this study, we have measured the singlet-oxygen quantum yield directly for the single-stranded T(T4tT)$_2$T$_2$ DNA oligonucleotide, demonstrating a 88% lower singlet-oxygen yield for T(T4tT)$_2$T$_2$ compared to that for 4tT under equal experimental conditions (Table 1). This result, together with the population of the triplet minimum of the thietane intermediate and the formation of the (6–4) photoadduct, establish that the primary photochemical mechanism of 4tT when incorporated into DNA is a [2 + 2] cycloaddition reaction instead of energy transfer to molecular oxygen to generate singlet molecular oxygen and other reactive oxygen species. Importantly, the (6–4) photoadduct is also expected to act as a Trojan horse[7,8], potentially leading to the formation of secondary cyclobutane pyrimidine dimer or oxidatively-generated damage of neighbor bases in 4tT-containing single- and double-stranded DNA.

In summary, this study presents the spectroscopic detection of a triplet-state minimum of the thietane intermediate, which inter-system crosses to populate the thietane intermediate in the ground state in a single-stranded DNA oligonucleotide. The timescale at which this triplet minimum is populated and the subsequent formation of the (6–4) photoadduct from the thietane intermediate in the ground state are determined. It further provides experimental evidence for the population of a triplet minimum of

the thietane intermediate in a time delay as short as ca. 3 ps directly from the triplet state of 4tT in the T(T4tT)$_2$T$_2$ single-stranded DNA oligonucleotide. The thietane intermediate in the ground state rearranges to form the (6–4) photoadduct, as evidenced from the irradiation experiments and from the determination of the characteristic absorption and fluorescence-emission spectra of this photoproduct. For complete disclosure, our results do not imply (or rule out) that the (6–4) photoadduct in the natural DNA or RNA system may also be formed from a charge-transfer state or from an initial electron transfer event. All these pathways could compete among each other in the formation of the (6–4) photoadduct, depending on the specific conformations, the redox properties, and electronic structure properties of the particular system under investigation (i.e., dinucleotides or single-versus double-stranded oligonucleotides, etc.).

Equally important, our results provide alternative insights for the investigation and understanding of the photodimerization mechanisms in dimeric sequences in single- and double-stranded DNA oligonucleotides containing thiobase analogs, where base sequence, stacking, and pairing interactions are expected to play a paramount role in controlling their photochemistry[50]. Additional experiments are currently underway in our group to investigate this chemistry in single- and double-stranded oligonucleotides.

## Methods

**Materials and steady-state spectroscopy**. 4-Thiothymidine (4tT, 99% purity) was obtained from Carbosynth Limited, Berkshire, UK, and was used as received. The HPLC-purified and lyophilized DNA oligonucleotide was obtained from the KareBay Biochem., Inc. Note that the incorporation of only one 4tT in a single-stranded oligonucleotide should be sufficient to form the thietane and the (6–4) photoproduct. Our choice of using an oligonucleotide containing two 4tT bases was made strategically to increase the absorbance of the oligonucleotide at the excitation wavelength of 342 nm relative to the absorbance at ca. 260 nm, where the thymine bases primarily absorb. This choice allows us to use less total amount oligonucleotide per experiment, and is also expected to increase the probability of thietane and (6–4) photoadduct formation, because up to two thietane could be formed per photoexcited oligonucleotide, hence, increasing the signal-to-noise ratio in the transient-absorption experiments.

Phosphate buffered saline (PBS) solutions with total phosphate concentration of 16 mM were freshly prepared using 0.4933, 0.4661, and 0.5048 g of monobasic sodium phosphate and 0.0653, 0.0961, and 0.0505 g of dibasic sodium phosphate dissolved in 250 mL of ultrapure water to generate PBS solutions of 7.4, 6.2, and 8.2, respectively. The pH was adjusted using 0.1 M solutions of NaOH and HCl. Steady-state absorption and emission spectroscopy were performed using a Cary 100 and Cary Eclipse spectrometers, respectively. Fluorescence spectra were taken at PMT voltage of 820 V, with slit widths of 5 nm, averaging time of 1 s and scan rate of 30 nm min$^{-1}$.

**Singlet-oxygen phosphorescence measurements**. Nanosecond time-resolved luminescence spectroscopy was used to determine the $^1O_2$ quantum yield ($\Phi_\Delta$) for T(T4tT)$_2$T$_2$ and 4tT in TRIS-buffered D$_2$O solution, under both air- and O$_2$-saturated conditions. A Spectra Physics GCR-150-30 Nd:YAG laser (355 nm, 7 ns pulse width) was used as the excitation source. Singlet-oxygen phosphorescence decay traces were collected at 1270 nm using a modified Fluorolog-3 spectrometer (HORIBA, Jobin Yvon) with a NIR sensitive photomultiplier tube (H10330A-45, Hamamatsu) and stored on a digital oscilloscope (TDS 360, Tektronics). Solutions of T(T4tT)$_2$T$_2$, 4tT, and the phenalenone standard were prepared in Tris-buffered D$_2$O with matching absorbance of 0.3 at 355 nm in 1 × 1 cm path length quartz cuvettes. The O$_2$-saturated solutions were bubbled with molecular oxygen for 20 min. The photodegradation of the sample was not allowed to reach more than 3%, as judged using UV–vis spectroscopy. The quantum yields were determined in back-to-back luminescence experiments of T(T4tT)$_2$T$_2$, 4tT, and phenalenone solutions under equal conditions, using the reported $\Phi_\Delta$ for phenalenone ($\Phi_\Delta = 0.98$)[52].

**Ultrafast broadband transient-absorption spectroscopy**. The broadband transient-absorption spectrophotometer made use of a Ti:Sapphire oscillator (Vitesse, Coherent, Santa Clara, CA, USA) that seeds a regenerative amplifier (Coherent Libra-HE) generating 100 fs pulses at 800 nm with a repetition rate of 1 kHz. The fundamental beam is used to pump a Traveling Optical Parametric Amplifier of Superfluorescence (TOPAS, Quantronix/Light Conversion, Vilnius, Lithuania), which in this work, was tuned to 342 nm. A translating 2 mm CaF$_2$ crystal was used to generate the white light continuum for probing. The absorbance of the oligonucleotide solution at the excitation wavelength of 342 nm was ~1.2,

while that of the 4tT monomer was 0.7. A 2-mm optical cell was used. The solutions were stirred continuously with a Teflon-coated magnetic stirrer. All solutions were kept below 10% photodegradation, as evaluated using UV–vis spectroscopy at the lowest-energy absorption band. Data processing made use of a home-made LabView program, while the target and global analyses were performed using the Glotaran software[64,65]. The decay traces obtained from the femtosecond transient-absorption data were globally fitted using a four-component sequential kinetic model, in conjunction with a coherent spectral fitting component to model the stimulated Raman emission, convoluted with a Gaussian instrument response function of $200 \pm 50$ fs (FWHM). The fourth kinetic component required a large lifetime (i.e., >3 ns) to fit the long-lived transient signals satisfactorily for both the oligonucleotide and the monomer, with the assumption that the long-lived signal decay exponentially.

**Determination of relative quantum yields.** Equation 1 was used to estimated the relative quantum yield of the triplet minimum of the thietane intermediate using the amplitudes from the global fit analysis (i.e., taken from the EADS) at the probe wavelength of 430 nm, as follows:

$$\text{relative \% quantum yield} = \frac{|A_n|}{|A_1| + |A_2| + |A_3| + |A_4|} \times 100, \qquad (1)$$

where $A_n$ are the amplitudes associated with the population of the triplet-state minimum of the thietane ($A_3 + A_4$) and the other amplitudes are associated with stimulated emission, triplet-state absorption and residual triplet-state absorption. Note that we used $A_3$ (430 nm) + $A_4$(430 nm) because both amplitudes contribute to the rise of the absorption band of the triplet minimum of the thietane at 430 nm during the second and third lifetimes, as can be observed in Supplementary Fig. 2b. Using this methodology, an average quantum yield value of $25 \pm 5\%$ was determined from two independent transient-absorption measurements. Note that this value should be considered as an upper limit because the triplet state of the 4tT in the oligonucleotide also absorbs at this probe wavelength. The reported uncertainty is twice the standard deviation.

Equation 1 was also used to estimate the fraction of the triplet state of 4tT that decays by self-quenching back to the ground state in the monomer at the probe wavelength of 550 nm, using the $A_3$ (550 nm) as the nominator ($30 \pm 5\%$).

**Density functional theory calculations.** All computations were performed using Gaussian 16 suite of programs[66]. The absorption spectrum of the triple-state minimum of the thietane and the thietane intermediate in the ground state were calculated using two different functionals, M052X and ωB97XD, and the 6-311 + +G(d,p) standard basis set. The absorption spectrum, and the vertical excitation energies and oscillator strengths of $T_{1CC}$ thietane triplet minimum, where calculated using the unrestricted ωB97XD functional and are reported in Supplementary Fig. 11 and Supplementary Table 2 for completeness. The optimized structures were taken from the work of Xie et al.[23]. Bulk solvent effects were modeled by using the integral equation formalism version of the polarizable continuum model[55].

## Data availability
The data supporting the findings of this study are available within the article and in its Supplementary Information. Additional data are available from the corresponding author upon reasonable request.

## Code availability
The software used in this study for global and target analyses is freely available at http://glotaran.org/.

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

## Acknowledgements

The authors acknowledge the National Science Foundation (Grant No. CHE-1800052). They also thank Mr. Brennan Ashwood for performing preliminary analysis of the transient-absorption data collected in earlier experiments. C.R. thanks the Deutsche Forschungsgemeinschaft (DFG) for support. This work made use of the High Performance Computing Resource in the Core Facility for Advanced Research Computing at CWRU.

## Author contributions

C.E.C.-H. conceived and designed the experiments; L.A.O.-R., C.R., S. H., and S.J. performed the experiments; L.A.O.-R., S. H., and S.J. analyzed the data; C.E.C.-H. and S.J. contributed reagents/materials/analysis tools; L.A.O.-R. and C.E.C.-H. wrote the paper. All authors read and approved the final version of the paper.

## Competing interests

The authors declare no competing interests.
