## [Peer Review File · Nature Communications]

Reviewers' comments:

Reviewer #1 (Remarks to the Author):

This interesting work brings new results dealing with the important issue of the (6-4) photoproduct formation through an elusive 4-membered ring compound. The proposed oxetane and azetidine intermediates formed during native DNA irradiation are too short-lived to be fully characterized, opening the door to the design of models to determine the excited states involved in their formation. In this context, 4-thiothymine has been widely considered as it forms a thietane with an adjacent thymine, which is stable enough to be characterized. However, to the reviewer knowledge, no clear conclusion has been drawn on the excited state responsible for this photocycloaddition. Here, the authors performed an elegant spectroscopic study of a short oligonucleotide containing two 4-thiothymine residues and monitored the phototriggered formation of a thietane derivative together with that of (6-4) photoproduct. This article should be of interest to a broad readership and deserves publication in Nature Commun. after taking into account the following main comments:

- 1- Although incorporation of only one thiothymine is enough to obtain the thietane and derived (6-4) photoproduct, the authors have used an oligonucleotide containing 2 thiobases separated by 1 thymine. ¿Why? No mention is made in the text to explain the choice of this particular sequence.
- 2- Some structures are needed to visualize thiothymine, thietane and (6-4) photoproduct.
- 3- Page 10. The transient species with a maximum at 430 nm was assigned to the thietane intermediate. This result is in agreement with the theoretical calculation run by the authors, showing band at 430 and 650 nm. A quantum of ca. 25% was estimated for the formation of this compound; however, the data (or methodology) used to determine this value have not been clearly specified in the main text or in the experimental section.
- 4- It has been previously described that the thietane is in equilibrium with its corresponding (6-4) photoproduct (see ref 11 and 14), the former being the predominant species. However, the yield of the latter increases under basic conditions (see reference 11). Here, formation of the (6-4) photoproduct is proposed to take place during the experiment performed at neutral pH. This is based on the transient absorption traces and kinetics as well as difference UV-absorption spectrum and steady-state fluorescence emission. Comparison of these data with those obtained at a more basic pH would allow displacement of the thietan/(6-4) photoproduct equilibrium, strengthening the conclusions given by the authors.
- 5- As a minor remark, p. 13, “establish that the primary photosensitization mechanism”. The term “photosensitization” is not appropriate as 4tT is a reactant here, “photochemical” should be better.

Reviewer #3 (Remarks to the Author):

This manuscript examines the early photochemistry that eventually leads to formation of 6-4 photoadducts in DNA using transient absorption and fluorescence spectroscopy. The main conclusion is that there is significant triplet quenching of 4tT upon incorporation into a 7-mer oligonucleotide and that this is related to the formation of the thietane intermediate. The authors have a strong history in this area and the mechanism described in the manuscript is both reasonable and seems to be consistent with the broader body of evidence available from the literature.

My complaint about the manuscript is that some of the new evidence from the transient absorption spectroscopy is not very convincing. The proposed model is mostly explained in terms of the small changes of the two bands at 430 nm and 560 nm in figure 3. However, these changes are very small and not very clearly seen in figures 3 and 4. The spectra in panels 3b and 3e are the most important for the discussed mechanism, but the variation between the cuts in these two panels seems to be about the same order magnitude as the fluctuation between points in the time-delay scans in figure 4 (also S5 and S6). It leaves me wondering how much of the spectral change is real or if it is mostly an artifact of the particular time cuts shown. The choice of time delays in panels 3b and 3e (and also the very limited number of spectra shown) is not very helpful either. According to figure 4b, the small rise at 430 nm is complete in only a few ps, so it's confusing that the time range in panel 3e extends out to 65 ps. This is important, because a similar rise is apparent in the 417 nm data of figure 4a, but is not evident in panel 3b because the signal in that case subsequently decays within the time window shown. In other words the time evolution is not very clear in figure 3 because of the limited number of cuts and the poor choice of time range. With so few points even the isosbestic points are not very convincing, especially for 4tT. The two panels really need to show more cuts and a time window up to only ~4ps, but I suspect this will further highlight the problems with the fluctuating signals which also needs to be addressed.

Another problem along the same lines is the comparison with calculations on page 11. The authors claim the calculated spectrum of thietane (Fig S4) matches the rise of the experimental transient absorption signal at 430nm. This is true, but the calculations show a much stronger band at ~630 nm that has no comparable signature in the experimental spectrum. How do they explain this important discrepancy?

Overall, the authors use some fairly strong language about the importance of their results that is probably not warranted by the data presented. Although I don't necessarily think the authors are wrong in their interpretations, the evidence is just not very compelling. The strength of this work is that it fits nicely into a broader context and therefore will be of interest to the community.

Minor points:

Page 7 - it's unclear what three bands are being discussed here. The authors should give wavelength ranges to clarify, and also comment explicitly about scattered pump light at 340 nm as a possible contamination of the signal on the short wavelength side of the spectrum.

Page 3 - the transition at 266 nm must be S0-Sn (not S1-Sn)

Page 4 - to avoid confusion, don't use a hyphen after O2

Page 8-9 (and elsewhere) - it should be 'single-stranded', not 'singlet-stranded'

Reviewer #4 (Remarks to the Author):

The submitted manuscript describes an experimental (stationary and transient absorption spectroscopy) study addressing the photochemistry of a single-strand DNA oligonucleotide including seven thymidines. The model employed contains a central 4-thiothymidine as the reactive center which makes possible to follow the photo-induced nucleotide reaction and associated mechanism as a function of time. The results are expected to elucidate the formation of the well known (6-4) DNA lesion which is one of the most frequent DNA lesion occurring at a single-strand level. The topic is of wide interest for the scientific community and, therefore, suitable for publication in Nat. Comm. On the other hand, the manuscript has to be greatly improved before publication can be recommended. In the following, I provide some general comments and few specific points that I hope will help the authors in revising the manuscript:

General comments.

First of all it has to be clarified that, as the authors correctly point out in the text, the theoretical/computational work that has inspired the research and provided the mechanism to be experimentally tested, is published (see ref. 29). While this is fine, the sentence ... “steady state and time-resolved spectroscopy are combined with quantum-chemical calculations...” in the Abstract suggests that the present work presents an equal amount of original experimental and computational results and this is not the case.

Secondly, this reviewer believes that the addressed problem (even when considering the specific thiothymidine-containing oligonucleotide model studied), requires a summary of the previously reported results and proposed mechanisms. It is not enough to write on pag. 2 (at the end of the second paragraph) that on the basis of different computational investigations, singlet and triplet $n\text{-}\pi^*$ and triplet $\pi\text{-}\pi^*$ have been proposed as precursor states for the formation of thietane. In other words the readership has to be informed on the background of the research and why the present study is superior.

Furthermore, as the author stress, the oxetane intermediate - which is the one that could formed naturally - has not been conclusively observed in DNA (Improta and Markovitsi published some joined work on the subject). Is there the possibility that a sulfur atom does not truly reflect what happen in the natural system? This question is, of course, central if one has to judge the impact of the presented research effort. I believe that a discussion of this point is given in some of the references. The work from Domratcheva (DOI:10.1002/chem.201700045) where I critical discussion of different aspects of the problem can be found including the formation of the triplet state. Also, what is the viewpoint of the authors of ref. 25 and 26?

Ref. 29 claims, essentially, formation of the triplet state of the S-substituted base. This triplet would drive formation of a triplet C-C intermediate (via a barrier) which then should somehow undergo another intersystem crossing and form the C-S bond. However, one may wonder if upon intersystem crossing to the ground single state and leading to the thietane, one also have a large amount of C-C fragmentation. Do the author see this process as well (that would decrease the quantum yield for the formation of the four-member ring)? It is critical to comment on the molecular structure corresponding to the second intersystem crossing.

The big merit of the manuscript remains the experimental verification (corroborated by the spectral band calculations at the TDDFT level) of extremely fast sequential intersystem crossings leading to a putative thietane intermediate in the ground state. As the authors stress this finding combined with the results of ref. 29 represent the main result of their work. Still it is not explained what is the main difference between the calculations in ref. 29 and those of the other theoretical contributions also suggesting the involvement of a triplet pathway.

(I understand that there are length restrictions in Nat. Comm. but I believe that without taking care of the points below in the main manuscript, the readership will not be satisfactorily served. This also include the specific point regarding the figures.

Specific points

1) One of the weaknesses of the manuscript is the lack of illustrations. One scheme with the relevant chemical formulas and one figure illustrating the final mechanism proposed have to be added to the main text. This is needed to facilitate the reading of the manuscript which must target the general scientific audience of Nat. Comm.

2) Please avoid terms like “high-level quantum-chemical calculations” or similar throughout the manuscript. They have little meaning. Rather, the authors should specify the type of quantum chemical method used.

3) Page 13. “instead of a Type II photosensitization mechanism”. Please, explain in the text, briefly, what such mechanism would involve (this point is obviously connected with some of the general points above).

Detailed Answers to Reviewers' Comments

Reviewer 1

General Comment: This interesting work brings new results dealing with the important issue of the (6-4) photoproduct formation through an elusive 4-membered ring compound. The proposed oxetane and azetidine intermediates formed during native DNA irradiation are too short-lived to be fully characterized, opening the door to the design of models to determine the excited states involved in their formation. In this context, 4-thiothymine has been widely considered as it forms a thietane with an adjacent thymine, which is stable enough to be characterized. However, to the reviewer knowledge, no clear conclusion has been drawn on the excited state responsible for this photocycloaddition. Here, the authors performed an elegant spectroscopic study of a short oligonucleotide containing two 4-thiothymine residues and monitored the phototriggered formation of a thietane derivative together with that of (6-4) photoproduct. This article should be of interest to a broad readership and deserves publication in *Nature Commun.* after taking into account the following main comments:

Reply: We thank the reviewer for carefully reading our manuscript and for providing comments that have assisted us to increase the clarity and overall impact of our work. We are also pleased that he/she finds the work of high significance and suitable for publication in *Nature Communications*.

Comment #1: Although incorporation of only one thiothymine is enough to obtain the thietane and derived (6-4) photoproduct, the authors have used an oligonucleotide containing 2 thiobases separated by 1 thymine. ¿Why? No mention is made in the text to explain the choice of this particular sequence.

Reply #1: We agree with the reviewer that the incorporation of only one 4-thiothymine should be sufficient to form the thietane and the (6-4) photoproduct. Our choice of using an oligonucleotide containing two 4-thiothymine bases was made strategically to increase the relative absorbance of the oligonucleotide at the excitation wavelength of 340 nm relative to the absorbance at ca. 260 nm, where the thymine bases primarily absorb. This choice allows us to use less total amount oligonucleotide per experiment (i.e., absorbance units) since the company sells the oligonucleotide per absorbance unit at 260 nm, not were the 4-thiothimide absorbs. Our choice of two 4-thiothimidine bases per oligonucleotide is also expected to increase the probability of thietane formation, since up to two thietane may be formed per photoexcited oligonucleotide, therefore, increasing the signal-to-noise ratio in the transient absorption experiments. This information is now included in the methods section of the revised manuscript.

Comment #2: Some structures are needed to visualize thiothymine, thietane and (6-4) photoproduct.

Reply #2: We agree with the reviewer. Two schemes showing the relevant structures are now included in the revised manuscript (i.e., Schemes 1 and 2).

Comment #3: Page 10. The transient species with a maximum at 430 nm was assigned to the thietane intermediate. This result is in agreement with the theoretical calculation run by the authors, showing band at 430 and 650 nm. A quantum of ca. 25% was estimated for the formation of this compound; however, the data (or methodology) used to determine this value have not been clearly specified in the main text or in the experimental section.

Reply #3: We thanks the reviewer for noticing this oversight. The relative percentage quantum yield of the triplet minimum of the thietane intermediate was estimated using the amplitudes from the global fit analysis (i.e., taken from the EADS) at the probe wavelength at 430 nm, as follows:

$$\text{relative \% quantum yield} = \frac{|A_n|}{|A_1| + |A_2| + |A_3| + |A_4|} \times 100 \quad (1)$$

where A_n are the amplitudes associated with the population of the triplet-state minimum of the thietane ($A_3 + A_4$) and the other amplitudes are associated with stimulated emission, triplet state absorption and residual triplet state absorption. Note that we used $A_3 + A_4$ because both amplitudes are contributing to the absorption band of the triplet-state minimum of the thietane at 430 nm during the second and third lifetimes, as can be seen in Figure S2b. The average value from two independent transient absorption measurements is $25 \pm 5\%$. We note herein and in the revised manuscript that Warren et al. (ref. 37) reported a thietane yield of 10% in an oligonucleotide containing one 4tThd. This suggests that a large fraction of the thietane in the triplet minimum intersystem crosses to populate the thietane intermediate in the ground state, while another fraction is not conducive to (6-4) photoadduct formation. The methodology used to estimate the relative quantum yield is now included in the methods section.

Comment #4: It has been previously described that the thietane is in equilibrium with its corresponding (6-4) photoproduct (see ref 11 and 14), the former being the predominant species. However, the yield of the latter increases under basic conditions (see reference 11). Here, formation of the (6-4) photoproduct is proposed to take place during the experiment performed at neutral pH. This is based on the transient absorption traces and kinetics as well as difference UV-absorption spectrum and steady-state fluorescence emission. Comparison of these data with those obtained at a more basic pH would allow displacement of the thietane/(6-4) photoproduct equilibrium, strengthening the conclusions given by the authors.

Reply #4: We thank the reviewer for proposing the idea of increasing the pH of the solution to displace the thietane/(6-4) photoproduct equilibrium towards the formation of the (6-4) photoproduct. The figure shown below reports the emission spectra of an irradiated solution of the oligonucleotide at three different pHs with equal concentrations of irradiated oligonucleotide. The solution was irradiated at neutral pH and then 10 μ L aliquots were taken to prepare the irradiated solutions at the three different pHs. The emission spectrum of the irradiated oligonucleotide increases with an increase in pH. This is consistent with a displacement of the thietane/(6-4) photoproduct equilibrium towards the (6-4) photoproduct upon an increase in pH (refs. 11 and 14 of the original manuscript) and further supports our assignment. The excitation spectra recorded at the emission wavelength of 390 nm have a maximum at ca. 320 nm for both basic and neutral pH conditions are in good agreement with the absorption spectrum of the (6-4) photoproduct. At acidic pH, the excitation shifts slightly to the blue to ca. 316 nm. Collectively, these results further support our main conclusions in the manuscript. The results are now included as supporting information and discussed in the revised version of the manuscript.

(a) Emission spectra of the irradiated oligonucleotide at different pHs; (b) Excitation Spectra of the irradiated oligo ($\lambda_{em} = 390$ nm).

Comment #5: As a minor remark, p. 13, “establish that the primary photosensitization mechanism”. The term “photosensitization” is not appropriate as 4tT is a reactant here, “photochemical” should be better.

Reply #5: We thank the reviewer for bringing this to our attention. We have revised the sentence accordingly.

Reviewer: 3

General Comment: This manuscript examines the early photochemistry that eventually leads to formation of 6-4 photoadducts in DNA using transient absorption and fluorescence spectroscopy. The main conclusion is that there is significant triplet quenching of 4tT upon incorporation into a 7-mer oligonucleotide and that this is related to the formation of the thietane intermediate. The authors have a strong history in this area and the mechanism described in the manuscript is both reasonable and seems to be consistent with the broader body of evidence available from the literature.

Reply: We thank the reviewer for carefully reading our manuscript and for providing comments that have assisted us to increase the clarity and overall impact of our work. We are also pleased that the reviewer asserts that the mechanism described in our work is reasonable and consistent with the broader body of evidence available in the literature.

Comment #1: My complaint about the manuscript is that some of the new evidence from the transient absorption spectroscopy is not very convincing. The proposed model is mostly explained in terms of the small changes of the two bands at 430 nm and 560 nm in figure 3. However, these changes are very small and not very clearly seen in figures 3 and 4. The spectra in panels 3b and 3e are the most important for the discussed mechanism, but the variation between the cuts in these two panels seems to be about the same order magnitude as the fluctuation between points in the time-delay scans in figure 4 (also S5 and S6). It leaves me wondering how much of the spectral change is real or if it is mostly an artifact of the particular time cuts shown. The choice of time delays in panels 3b and 3e (and also the very limited number of spectra shown) is not very helpful either. According to figure 4b, the small rise at 430 nm is complete in only a few ps, so it's confusing that the time range in panel 3e extends out to 65 ps. This is important, because a similar rise is apparent in the 417 nm data of figure 4a, but is not evident in panel 3b because the signal in that case subsequently decays within the time window shown. In other words, the time evolution is not very clear in figure 3 because of the limited number of cuts and the poor choice of time range. With so few points even the isosbestic points are not very convincing, especially for 4tT. The two panels really need to show more cuts and a time window up to only ~4ps, but I suspect this will further highlight the problems with the fluctuating signals which also needs to be addressed.

Reply #1: We understand the reviewer's concerns. Therefore, we decided to perform the transient absorption experiments two additional times in two separate days, each day weeks apart from each other and using two different batches of oligonucleotides (this time from KareBay Biochem, Inc.), to verify the reproducibility of our measurements. Accomplishing these additional experiments was the bottleneck in submitting the revised manuscript. Firstly, the company that we initially bought the oligonucleotide (Midland Co.) informed us that they were unable to make the oligonucleotide for us (after two months of waiting time) because they had too many other orders to complete before working with ours. We then contacted KareBay and they agree to make the oligonucleotide for us, but it took about 1.5 months to prepare the oligonucleotide each time that we requested a new batch of the oligonucleotide. Even though this process significantly delayed the resubmission of this manuscript, we felt that following this rigorous procedure was necessary in order to answer the concern raised by the reviewer and to be confident on the reproducibility of the work.

It is gratifying to report that the results for the oligonucleotide are not only qualitatively, but quantitatively reproducible. We were also able to improve the signal-to-noise ratio of the data significantly, as can be observed in new Figures 3 and 4 of the revised manuscript and Figures S3, S7, and S8. The transient data for the 4tThd were also reproducible, but the putative isosbestic point was not clearly observed in the two separate data sets collected for 4tThd in back-to-back with the new oligonucleotide data. This is in complete agreement with previous observations for 4tThd reported by our group (as suspected by the reviewer). We thank the reviewer for motivating us to repeat the transient absorption experiments for both 4tT and the oligonucleotide.

We agree that the rise in the absorption band around 430 nm is relative small. However, this is not surprising because it has been shown by Warren et al. (ref. 14) that the quantum yield of (6-4) photoproduct formation is small in oligonucleotides containing one 4tT (ca. 10%). In our view, the relatively small rise in absorption for the 430 nm band does not diminish the significance of the results. To the contrary, we believe they just highlight the challenges involved in performing these experiments due to the relatively small quantum yield of thietane intermediate formation. Furthermore, the transient absorption results for the oligonucleotide are clearly different from those taken for the 4tT in back-to-back experiments under equal experimental conditions. Specifically, while the transient absorption spectra of 4tT decay in the spectral probe region from ca. 350 to 670 nm, those of the oligonucleotides clearly increase from ca. 380 to 450 nm, while they decay from ca. 450 to 670 nm. The visible band assigned to the triplet state of 4tT in the oligonucleotide decays significantly faster than in the monomer. This is further supported by the global analysis results and by the representative decay traces shown in Figures 4 (and Figures S7 and S8). In the revised Figure 3, we have now added additional transient absorption spectra at different time delays, as requested by the reviewer.

Comment #2: Another problem along the same lines is the comparison with calculations on page 11. The authors claim the calculated spectrum of thietane (Fig S4) matches the rise of the experimental transient absorption signal at 430nm. This is true, but the calculations show a much stronger band at ~630 nm that has no comparable signature in the experimental spectrum. How do they explain this important discrepancy?

Reply #2: We thanks the reviewer for encouraging us to take a closer look at this apparent discrepancy. Of course, we were aware of the differences between the calculated spectrum and the transient spectra. In the original manuscript, we proposed that this discrepancy could have originated from significantly different absorption cross sections between the absorption spectra of the triplet state and the thietane in this spectral region. In other words, the triplet state of the 4tT in the oligonucleotide absorbs significantly more strongly in the visible than the thietane, while the thietane absorbs significantly more at 430 nm than the triplet state. This was deemed as a reasonable explanation, because we do not know the excited-state absorption cross sections for each of these species.

However, we decided to recalculate the absorption spectrum for this triplet minimum of the thietane intermediate (labeled T_{1CC} , in Xie et al. paper, ref. 29) and in doing so realized that we had used a singlet character instead of a triplet character, as reported by Xie et al. in their theoretical paper (ref. 29). As shown in the revised Fig. S6 of the revised ESI (see figure below), the absorption spectrum of this triplet structure has a large absorbance around 470 and 350 nm with barely no absorption around 550 nm. The revised absorption spectrum for this triplet intermediate is in even better agreement with the transient results, further supporting the assignment of the transient species to the T_{1CC} state species.

Figure S6. Calculated absorption spectrum of T_{10C} in water at the TD-UM052X/IEFPCM/6-311++G(d,p) level of theory. In order to plot the results obtained, each transition was convoluted by a Gaussian with a FWHM of 0.33 eV. The optimized structure of the T_{10C} was taken from a recent work published by Xie et al. (ref. 29), which was calculated at the QM(CASSCF)/MM level of theory.

Comment #3: Overall, the authors use some fairly strong language about the importance of their results that is probably not warranted by the data presented. Although I don't necessarily think the authors are wrong in their interpretations, the evidence is just not very compelling. The strength of this work is that it fits nicely into a broader context and therefore will be of interest to the community.

Reply #3: We are not quite sure what the reviewer means by "some fairly strong language" regarding the relevance of our results. However, in the revised version, we have tuned-down generalizations and clarified the significance of our work to the natural DNA systems.

Comment #4: Minor points:

Page 7 - it's unclear what three bands are being discussed here. The authors should give wavelength ranges to clarify, and also comment explicitly about scattered pump light at 340 nm as a possible contamination of the signal on the short wavelength side of the spectrum.

Page 3 - the transition at 266 nm must be S0-Sn (not S1-Sn)

Page 4 - to avoid confusion, don't use a hyphen after O2

Page 8-9 (and elsewhere) - it should be 'single-stranded', not 'singlet-stranded'

Reply #4: We thank the reviewer for bringing these issues to our attention. They are all corrected in the revised version of the manuscript.

Reviewer: 4

General Comment: The submitted manuscript describes an experimental (stationary and transient absorption spectroscopy) study addressing the photochemistry of a single-strand DNA oligonucleotide including seven thymidines. The model employed contains a central 4-thiothymidine as the reactive center which makes possible to follow the photo-induced nucleotide reaction and associated mechanism as a function of time. The results are expected to elucidate the formation of the well known (6-4) DNA lesion which is one of the most frequent DNA lesion occurring at a single-strand level. The topic is of wide interest for the scientific community and, therefore, suitable for publication in *Nat. Comm.*

Reply: We thank the reviewer for carefully reading our manuscript and for highlighting the fact that the topic is of wide interest for the scientific community and the work suitable for publication in *Nature Communications*. We would also like to thank the reviewer for providing excellent comments that have allowed us to improve the manuscript.

Comment #1: On the other hand, the manuscript has to be greatly improved before publication can be recommended. In the following, I provide some general comments and few specific points that I hope will help the authors in revising the manuscript:

General comments.

First of all it has to be clarified that, as the authors correctly point out in the text, the theoretical/computational work that has inspired the research and provided the mechanism to be experimentally tested, is published (see ref. 29). While this is fine, the sentence ... "steady state and time-resolved spectroscopy are combined with quantum-chemical calculations..." in the Abstract suggests that the present work presents an equal amount of original experimental and computational results and this is not the case.

Reply #1: Our intention in the Abstract is highlighting the fact that we use ... "steady state and time-resolved spectroscopy are combined with quantum-chemical calculations...", which is exactly what we did, not to suggest that our work present an equal amount of experimental and computational results. We believe that it is important to highlight that we have used both experimental and computational results to arrive at the main conclusions of our work. This is so because it would be very difficult to arrive to such conclusions in the first place without calculating the absorption spectra of the triplet-minimum of the thietane and of the thietane intermediate in the ground state and comparing it with the transient absorption spectra. Furthermore, we believe that the manuscript itself clearly displays the amount of calculations that we have done and, equally important, provides the necessary credit to the authors of the original computational work that inspired our experiments.

Comment #2: Secondly, this reviewer believes that the addressed problem (even when considering the specific thiothymidine-containing oligonucleotide model studied), requires a summary of the previously reported results and proposed mechanisms. It is not enough to write on pag. 2 (at the end of the second paragraph) that on the basis of different computational investigations, singlet and triplet $n\text{-}\pi^*$ and triplet $\pi\text{-}\pi^*$ have been proposed as precursor states for the formation of thietane. In other words the readership has to be informed on the background of the research and why the present study is superior.

Reply #2: We agree with the review that the readership of our manuscript can benefit from a brief summary of the previous work and proposed mechanisms. Therefore, we have included Scheme 1 below and revised the Introduction section accordingly.

Scheme 1. Proposed excited-state precursor in the literature for the formation of the oxetane intermediate.^{2-5,10-22}

Comment #3: Furthermore, as the author stress, the oxetane intermediate - which is the one that could formed naturally - has not been conclusively observed in DNA (Improta and Markovitsi published some joined work on the subject). Is there the possibility that a sulfur atom does not truly reflect what happen in the natural system? This question is, of course, central if one has to judge the impact of the presented research effort. I believe that a discussion of this point is given in some of the references. The work from Domratcheva (DOI:10.1002/chem.201700045) where I critical discussion of different aspects of the problem can be found including the formation of the triplet state. Also, what is the viewpoint of the authors of ref. 25 and 26?

Reply #3: We thank the reviewer for bringing to our attention the paper by Domratcheva and co-workers. With regard to the question as to whether the use of a 4tT-containing oligonucleotide is a good model system to investigate the formation of the (6-4) photoproduct in natural DNA systems, we argue, as many others have done before us, that it is in fact an excellent system. What we are able to demonstrate with this system is that the triplet state gives rise to the thietane formation, which in turn give rise to the (6-4) photoproduct between 4tT and thymidine (Thd) in a DNA oligonucleotide. This model system lend strong support to the earlier experimental and theoretical proposals that the excited triplet state of Thd can give rise to the formation of the (6-4) photoproduct in natural DNA, probably following a similar mechanism than the one we propose herein for the reaction between 4tT and Thd. This is clearly an important result of direct relevance to the natural system. However, to be completely clear, our results do not imply (or rule out) that the (6-4) photoproduct in the natural system could also be formed from charge transfer states, or from an initial electron transfer event. All these pathways may compete among each other in the formation of the (6-4) photoadduct in DNA, depending on the conformation(s), the redox properties, and electronic structure of the particular system under studied (i.e., dinucleotide, single- versus double-stranded oligonucleotides, etc.). A note has been added to the manuscript to highlight this fact.

For example, the work of Domratcheva provides compelling evidence for the formation of a (6-4) photoproduct analogue from the initial electron transfer from a xylene moiety to Thd. However, this is not surprising given the significant differences among the redox potential of these two molecules. What we think that is particularly relevant from this work, with regards to a natural DNA dinucleotide, is the demonstration that the directionality of the reaction partners play an important role in the formation of the (6-4) photoproduct analogue, as has been observed in the natural system (and also in 4tT-containing oligonucleotides). We also note that the work by Domratcheva and co-workers also provides important mechanistic insights for understanding the mechanism of formation of (6-4) photoproduct in the natural system based on quantum-mechanical calculations. It is possible, however, that the formation of the (6-4) photoproduct between two Thd in natural DNA may also occur from a charge transfer state

between the 5'-Thd to the 3'-Thd or from a triplet state with significant charge transfer character, as other computational works have predicted. The experimental results obtained by Domratcheva et al. and us using model DNA systems demonstrate that both electron transfer and population of the triplet state can give rise to the formation of the (6-4) photoproduct. These experimental observations are quite relevant, however, neither work can show which pathway is actually the most competitive one in a natural DNA system.

Comment #4: Ref. 29 claims, essentially, formation of the triplet state of the S-substituted base. This triplet would drive formation of a triplet C-C intermediate (via a barrier) which then should somehow undergo another intersystem crossing and form the C-S bond. However, one may wonder if upon intersystem crossing to the ground single state and leading to the thietane, one also have a large amount of C-C fragmentation. Do the author see this process as well (that would decrease the quantum yield for the formation of the four-member ring)? It is critical to comment on the molecular structure corresponding to the second intersystem crossing.

Reply #4: We cannot provide experimental evidence to support or refute the putative fragmentation of the C-C bond in the ground state before the thietane is formed. We note, however, that the photoreactivity of oligonucleotides containing 4tT is relatively small (ca. 10%) according to the work of Warren et al. (ref. 14). In our femtosecond transient absorption experiments, we unequivocally detect the ultrafast population of the triplet state of 4tT, which is followed by a branching process to populate the triplet state minimum of the thietane intermediate (labeled T_{10C} by Xie et al. in ref. 29), or the ground state of 4tT. According to the calculations of Xie et al, the triplet minimum of thietane intersystem crosses to the ground state of thietane before thermally decomposing to form the (6-4) photoproduct. This is now explained in significant detail in the revised manuscript, as requested by the reviewer.

Comment #5: The big merit of the manuscript remains the experimental verification (corroborated by the spectral band calculations at the TDDFT level) of extremely fast sequential intersystem crossings leading to a putative thietane intermediate in the ground state. As the authors stress this finding combined with the results of ref. 29 represent the main result of their work. Still it is not explained what is the main difference between the calculations in ref. 29 and those of the other theoretical contributions also suggesting the involvement of a triplet pathway.

(I understand that there are length restrictions in Nat. Comm. but I believe that without taking care of the points below in the main manuscript, the readership will not be satisfactorily served. This also include the specific point regarding the figures.

Reply #5: Because different theoretical groups use different levels of theory (DFT, CASSPT2, CASSCF, QM-MM, etc.), different systems (free thymine bases, dinucleotides, single- or double-stranded DNA, etc.), and different solvent conditions (i.e., vacuum, reaction field solvation model, explicit water molecules, etc.) it is impossible to compare them in any meaningful way under restricted page limits. We also think that such technical and specialized comparisons will not benefit significantly the broader readership of *Nature Communications*.

Comment #6: Specific points

1) One of the weaknesses of the manuscript is the lack of illustrations. One scheme with the relevant chemical formulas and one figure illustrating the final mechanism proposed have to be added to the main text. This is needed to facilitate the reading of the manuscript which must target the general scientific audience of Nat. Comm.

Reply #6a: We agree with the reviewer that the manuscript can benefit from additional illustrations. We have added Scheme 1 and Scheme 2 to the revised manuscript. Scheme 1 depicts the different excited-state precursors

proposed in the literature for the formation of the oxetane intermediate, while Scheme 2 shows the proposed reaction mechanism leading to the formation of the (6-4) photoproduct analog.

2) Please avoid terms like “high-level quantum-chemical calculations” or similar throughout the manuscript. They have little meaning. Rather, the authors should specify the type of quantum chemical method used.

Reply #6b: We have now specified the quantum-chemical methods used by different authors whenever it is necessary to add clarity to our work.

3) Page 13. “instead of a Type II photosensitization mechanism”. Please, explain in the text, briefly, what such mechanism would involve (this point is obviously connected with some of the general points above).

Reply #6c: Done.

REVIEWERS' COMMENTS:

Reviewer #1 (Remarks to the Author):

The authors carefully addressed the reviewers' comments and the manuscript is much improved. Therefore, I support its publication in its current form.

Reviewer #3 (Remarks to the Author):

The authors have addressed reviewer comments and the manuscript is much better for it, including expanded background, clarification of the proposed mechanisms and additional discussion of the results that will be very helpful to the reader. I appreciate that the authors have gone to great effort to reproduce the experimental transient spectra and confirm the trends in figure 3. It's regrettable that the DNA samples were not more readily available but having confirmed the reproducibility of the results and clarified some points was worth the effort and strengthens the interpretation. This work will be of interest to a wide audience and offers new evidence for an interesting and important process.

Reviewer #4 (Remarks to the Author):

After a careful reading of the revised manuscript and of the reply to the points I raised in my past review, I find that no additional conceptual and/or technical improvement are necessary. Accordingly, I am now able to recommend the manuscript for publication in Nat. Comm.

Additional minor change.

1) Abstract: I ask the author to change the sentence:

"...combined with quantum-chemical calculations..." to "...combined with new and reported quantum-chemical calculations...".

Point-by-point response

REVIEWERS' COMMENTS

Reviewer #1 (Remarks to the Author):

The authors carefully addressed the reviewers' comments and the manuscript is much improved. Therefore, I support its publication in its current form.

Response: We thank the reviewer for his/her support.

Reviewer #3 (Remarks to the Author):

The authors have addressed reviewer comments and the manuscript is much better for it, including expanded background, clarification of the proposed mechanisms and additional discussion of the results that will be very helpful to the reader. I appreciate that the authors have gone to great effort to reproduce the experimental transient spectra and confirm the trends in figure 3. It's regrettable that the DNA samples were not more readily available but having confirmed the reproducibility of the results and clarified some points was worth the effort and strengthens the interpretation. This work will be of interest to a wide audience and offers new evidence for an interesting and important process.

Response: We fully agree with the reviewer and thank him/her for the comments that significantly enhanced the revised version of this manuscript.

Reviewer #4 (Remarks to the Author):

After a careful reading of the revised manuscript and of the reply to the points I raised in my past review, I find that no additional conceptual and/or technical improvement are necessary. Accordingly, I am now able to recommend the manuscript for publication in Nat. Comm.

Additional minor change.

1) Abstract: I ask the author to change the sentence:

"...combined with quantum-chemical calculations..." to "...combined with new and reported quantum-chemical calculations...".

Response: We thank the reviewer for the support. We have added the proposed phrase to the Abstract, as requested.